# Promoting or Inhibiting? The Impact of Environmental Regulation on Corporate Financial Performance—An Empirical Analysis Based on China

**DOI:** 10.3390/ijerph17113828

**Published:** 2020-05-28

**Authors:** Xiang Deng, Li Li

**Affiliations:** School of Economics, Sichuan University, No. 24 South Section 1 Yihuan Road, Chengdu 610065, China; dengxiang@scu.edu.cn

**Keywords:** environmental regulation, new environmental protection law, corporate financial performance, environmental supervision intensity

## Abstract

Today, environmental protection has become a global issue, and various environmental regulations have been actively adopted. However, are these measures promoting or harming enterprise values? Is this effect the same for enterprises with different ownership backgrounds? In order to address these problems, we conducted an empirical analysis of China’s A-share market to investigate the relationship between the New Environmental Protection Law (NEPL) launched in China and corporate financial performance, and further explore the impact of environmental supervision intensity (ESI) from the perspective of ownership. The empirical results show that there is a negative correlation between NEPL and the financial performance of high pollution enterprises. Further analysis demonstrates that there is an inverted U-shape relationship between ESI and corporate financial performance for both state-owned enterprises (SOEs) and non-state-owned enterprises (non-SOEs), while the financial performance of SOEs is more sensitive and tolerant to environmental regulation than that of non-SOEs. Finally, we make recommendations for the future direction of China’s ecological civilization construction and sustainable development of enterprises based on three aspects: environmental awareness, policy considerations, and sustainable development. The innovation of this paper lies in putting NEPL and corporate financial performance in the same analytical framework for the first time, which enriches the research in this field. Meanwhile, it provides a new perspective for understanding the relationship between ESI and corporate financial performance through the analysis of nonlinearity and owner heterogeneity.

## 1. Introduction

Environmental problems, such as pollution, resource depletion, and biodiversity destruction have increasingly become the focus of global concern [1,2,3]. Whether these problems can be properly solved is crucial to human survival and social development [4,5]. To solve the environmental problem properly, international organizations and countries around the world have made many efforts. As the largest international environmental organization, the United Nations Environment Programme (UNEP) has promoted a series of conventions, from the “United Nations Framework Convention on Climate Change” and the “Kyoto Protocol” to the “Paris Agreement” [6,7,8]. As the first country to establish a long-term legally binding framework to cut carbon emissions, the UK passed the Climate Change Act in 2008 [9]. Over the past decades, the European Union has also ratified many international environmental agreements, such as the Treaty of Amsterdam and the 7th Environment Action Programme, to curb air, water and soil pollution, among others [10]. Later, in 2019, the Japanese G20 Presidency hosted the first-ever G20 Ministerial Meeting on Environment, Climate Change and Energy, which also demonstrated the willingness of both developed and developing countries to work together to promote energy transition and sustainable global environmental growth.

As the second-largest economy in the world, China has also made significant contributions to solving its growing environmental problems, for example through the New Environmental Protection Law (NEPL) issued in 2015 and the central government’s environmental supervision action launched in 2016. Similarly, as a member of the international community, China, while making great efforts to protect its own environment, has also assumed an active role in international environmental affairs, striving to promote international cooperation in the field of environmental protection. On the eve of the Copenhagen Climate Change Conference, for instance, China pledged to cut carbon dioxide emissions per unit of gross domestic product by 40–45% by 2020 compared to 2005 levels, which aims to further demonstrate the determination of environmental governance.

However, protecting the environment and developing the economy seem to be contradictory goals [11,12]. Some scholars claimed that environmental regulation imposes various restrictions on enterprises’ emission behaviour, which inevitably leads to increases in costs and decreases in the profits of enterprises [13,14,15]. However, Porter and van der Linde argued that properly designed environmental standards could, to some extent, trigger innovation, offset the costs of compliance, and enhance competitiveness, which is called the Porter hypothesis (PH) [16,17,18].

After PH was put forward, scholars have conducted extensive studies on the relationship between environmental regulation and enterprise development. Three approaches emerge from this empirical literature. The first intends to analyze the link between environmental regulation and innovation. Operationally, innovation is generally defined through R&D expenses or the number of patents [19,20,21,22,23,24,25,26]. The results on the relationship varies significantly, with positive effects [21,22,23,24,25], negative effects [19], and no effects [20]. The second empirical approach assesses the impact of environmental regulation on the business performance of the firm. The firm’s business performance is often measured by its productivity [27,28,29,30,31,32]. Most papers highlight a negative impact of environmental regulation on productivity [27,31,32]. However, several studies reported more positive results [28,29]. For example, Berman and Bui found that, although Los Angeles implemented a stricter air pollution regulation, the oil refineries located in the city showed a significantly higher productivity than other U.S. refineries [28]. However, some scholars argued that the relationship is uncertain [30]. The third approach aims at exploring the relationship between corporate environment performance and economic performance [33,34,35,36,37,38,39].

As for the research concerning the environmental regulation intensity and enterprise performance, it can roughly be divided into two aspects. First, the relationship between policy stringency and net exports or competitiveness, which usually corresponds to the pollution haven hypothesis [40]. Much of the earlier literature found little evidence that stringent environmental regulations affect net export [15,41,42,43]. However, there is also evidence supporting the positive correlation [44,45]. Another aspect is the relationship between stringency and productivity or innovation [46,47]. In a 1983 article, Gollop found that strict environmental regulations can lead to a decline in business productivity [11]. However, recent research has found that tighter policies lead to short term gains in productivity [47].

Existing literature mainly focuses on the impact of environmental regulation on the productivity or innovation of enterprises, as well as the relationship between environmental management and economic performance, but rarely considers the direct impact of environmental regulation on corporate financial performance, in particular the effect of the NEPL. In addition, the inconsistencies in the research on environmental supervision intensity (ESI) may be attributed to methodological differences. In previous studies, researchers usually conducted regression analyses within a linear framework, which resulted in relevant discrepancies in the conclusions. However, the actual relationship between ESI and corporate financial performance may be more complex than simply positive, negative, or neutral.

As for the discussion of heterogeneity, the existing studies mainly emphasize the impact of industry heterogeneity and regional heterogeneity [48,49], and rarely analyze it from the perspective of ownership. However, previous studies indicate that the effect of policies may be affected by the heterogeneity of ownership, and enterprises with different ownership backgrounds may have different responses to policies [5,50,51]. Thus, ownership heterogeneity may also have a significant impact on companies’ compliance with environmental policies. In particular, in China, there is a remarkable difference between the social status of state-owned enterprises (SOEs) and non-state-owned enterprises (non-SOEs) in the whole national economy.

Therefore, in order to better understand the relationship between environmental regulation, namely the NEPL and ESI in this paper, and corporate financial performance, our paper starts from the perspective of ownership heterogeneity. In addition, considering the effect of changes in ESI on enterprise financial performance may be far more complex than purely linear, our paper explores it from a non-linear prospective, that is, by introducing the quadratic term of ESI. Thus, to evaluate the impact of environmental regulation on corporate financial performance and further investigate the heterogeneous effect under different ownership backgrounds, we use quarterly data of China’s A-share listed companies from 2012Q1 to 2019Q3 for the empirical analysis.

The contribution of this paper lies in the following three aspects. Firstly, this paper puts NEPL and corporate financial performance in the same analytical framework for the first time, which enriches the research in this area. Secondly, from a nonlinear perspective, a more detailed conclusion is drawn from the discussion of the economic impact of the changes in ESI. We find that ESI does not simply inhibit enterprise performance, but shows an inverted U-shape effect. Finally, a further study on the ESI finds that ownership heterogeneity matters in environmental regulation, that is, the financial performance of SOEs is more sensitive and tolerant to environmental regulation than that of non-SOEs in Chinese institutional environment. The conclusion of this paper provides a new perspective on how to formulate environmental policies to achieve sustainable economic development in the future.

The rest of this article is organized as follows. Section 2 outlines the theoretical analysis and research hypothesis. Section 3 describes the data resources, key variables and benchmark model. Section 4 outlines the empirical results of NEPL and ESI and conducts further analyses of heterogeneity. Section 5 offers some conclusions and policy recommendations.

## 2. Theoretical Analysis and Research Hypothesis

### 2.1. NEPL and Corporate Financial Performance

The NEPL was adopted on 24 April 2014 and came into effect on 1 January 2015. Compared with the old environmental protection law, the prominent feature of NEPL is that it not only strengthens the punishment of illegal polluters, but also highlights the environmental supervision responsibility of local governments [52]. On the one hand, the environmental regulatory authorities can directly seal up or detain the facilities and equipment, impose daily penalties as well as detain the persons directly in charge of the enterprises that violate the environmental laws and regulations, such as evading supervision. On the other hand, the NEPL emphasizes that environmental protection departments of government at all levels should be given certain punishments if they fail to effectively fulfil their law enforcement obligations, and environmental performance should be taken as an important consideration in performance appraisal of government officials. Therefore, the introduction of NEPL has greatly enhanced the effectiveness of environmental law enforcement.

As the major producers of environmental pollution, compared to low-polluting enterprises (LPEs), high-polluting enterprises (HPEs) have always been the focus of government regulators [53]. For the sake of political performance, local governments are bound to strictly implement environmental policies and restrict or stop the production of HPEs that violate the emission rules. If the circumstances are serious, they may even be ordered to suspend business or close down. In addition, HPEs also have to increase investment in environmental protection to cope with rising environmental standards. All this will inevitably lead to the decline of profits as well as their financial performance [27,54,55]. Hence, based on the above analysis, a hypothesis (H1) is proposed as follows:

**Hypothesis** **1.**
*The introduction of the NEPL will reduce the financial performance of HPEs.*


### 2.2. ESI and Corporate Financial Performance of HPEs and LPEs

At the beginning of 2016, the Chinese central environmental supervision group conducted a pilot inspection of environmental protection in the Hebei province, and the environmental supervision action officially kicked off. By the end of 2017, all 31 provinces of China had completed one inspection each. In May and October 2018, the central environmental supervision team conducted a “look back” operation in twenty provinces and regions in two batches, aiming at examining if the problems discovered during the first inspection had been solved. In July 2019, the second round of environmental supervision was officially launched. The first batch of eight inspection teams successively entered six provinces and cities, including Shanghai, Fujian, Hainan, Chongqing, Gansu and Qinghai, and two SOEs, namely China Minmetals Corporation (Beijing, China) and China National Chemical Corporation Ltd. (Beijing, China).

The 26 provinces in China that have gone through a retrospective and a second round of environmental supervision are bound to have more aggressive policy enforcement than the others; therefore, the financial performance of enterprises in these provinces is more likely to be affected. Meanwhile, studies have shown that regulation can disproportionately harm small businesses if firms of all sizes are treated equally [56,57,58]. Once environmental supervision begins, small and medium-sized enterprises (SMEs) are more likely to be targeted by law enforcement [59,60]. On the one hand, due to financial capital and human resource constraints, SMEs are unable to make substantial investments in environmental issues [61,62,63]. On the other hand, the production efficiency of high pollution SMEs is lower than that of large listed enterprises, and their relatively lower profits make them less motivated to invest in environmental protection [62,64]. Additionally, since SMEs lack awareness of environmental legislation [63,65], once environmental supervision increases in intensity, a large number of high pollution SMEs face either being closed or having their production stopped for rectification [53,66,67], the latter being just as detrimental to them as the former.

Therefore, given that the demand for the products produced by these enterprises remains high, the demand will flow to large listed HPEs, thus helping them recover from the loss caused by environmental supervision and improving their financial performance [67]. However, as environmental supervision intensifies further, listed HPEs will also inevitably be subject to stricter supervision, and many of them will be required to stop production or invest significantly in environment protection. Consequently, the performance of these enterprises will be greatly affected as well [53]. Therefore, when the intensity of environmental supervision is low, the financial performance of listed HPEs may improve as SMEs bear the brunt, and once it exceeds a certain level, their performance will also decline.

As for LPEs, when faced with environmental regulations, the competitive advantages of such enterprises will emerge. Because the products they produce are more environmental-friendly, their social reputation is generally higher, and a better reputation can help them win the favour of more investors [36,68], so their financial performance will be improved to some extent [55]. However, once the intensity of environmental supervision exceeds a certain threshold, these companies also have to pay considerable compliance costs to maintain the high-standard environmental protection rules. Therefore, the performance of LPEs will also decline due to the externality of public policy.

In conclusion, this paper argues that the relationship between ESI and corporate financial performance may not be a simple linear one. Based on the above analysis, a hypothesis (H2) is proposed as follows:

**Hypothesis** **2.**
*With the intensify of ESI, corporate financial performance for both listed HPEs and LPEs will increase at first and then decline, that is, the relationship between them is an inverted U-shape.*


### 2.3. Ownership Heterogeneity, ESI and Corporate Financial Performance

China’s economy is based on the co-ownership system and is developed by multiple forms of ownership. The difference in ownership, reflected in the political connections with the government, is the typical characteristic of Chinese enterprises. SOEs have a natural political connection with the government. As the main body of China’s economy, SOEs are not only an important guarantee of economic growth, but also endowed with social goals of ensuring employment and maintaining regional stability [69,70]. Many studies have shown that companies can benefit from political connections, such as more bank loans [71], tax breaks [72], and higher stock returns [73]. Although private enterprises are the source of China’s growth miracle [74], their regulatory burden is much greater than that of SOEs [75,76]. Much empirical evidence shows that SOEs have obvious advantages over non-SOEs in obtaining subsidies, bank credit, and other cheap factor inputs [77,78]. Also, recent research suggests that political connections may also help companies circumvent government environmental regulation [79]. The property right nature of SOEs strengthens the “shelter effect” of political connections on the punishment for environmental violations by enterprises [70].

Given that political connection can adjust the implementation intensity of environmental regulation policies in enterprises of different ownership, the real pressure on SOEs and non-SOEs may be different even in the face of the same environmental regulation intensity.

When the government strengthens environmental regulation, SOEs, by virtue of their close relationship with the government, are more convenient and unimpeded in the information transmission channels. They can timely and accurately grasp the government’s environmental attitude and intention, so as to respond quickly to environmental regulation, improve their financial performance and show greater sensitivity, that is, a steeper inverted U-shaped curve [55,80]. However, due to their lower political affiliation, non-SOEs bear significantly higher pressure than SOEs, and their tolerance limit of environmental regulation will be reached faster, which inevitably leads to an earlier reversal of their financial performance towards environmental regulation, that is, an earlier infection point. Hence, based on the above analysis, a hypothesis (H3) is proposed as follows:

**Hypothesis** **3.**
*With the intensify of ESI, the financial performance of SOEs is more sensitive and tolerant to environmental regulation than that of non-SOEs, that is, the inverted U-shaped curve is steeper, and the inflection point occurs later for SOEs.*


## 3. Benchmark Model and Data Resource

This section provides a description of the empirical data, main variables and econometric model. We construct an empirical model to examine the relationship between environmental regulation and enterprises’ financial performance via the difference in difference (DID) model by summarizing models on environmental regulation and policy evaluation [81,82,83]. To further analyze the impact of environmental regulation intensity on enterprises’ financial performance, we also introduce the fixed effects (FE) model for analysis.

### 3.1. Data Source and Sample Selection

The data of sample companies in this paper are derived from the WIND database, including the quarterly data of 3707 Chinese A-share listed companies from 2012Q1 to 2016Q2. We divide the samples into HPEs and LPEs according to the Environmental Protection Verification of Listed Companies Industry Classification Management List (No. 373 (2008)) listed by the Ministry of Environmental Protection in 2008. Based on the stability of the data and the economic meaning of variables, this paper deals with the data as follows: (1) based on previous research [54,55,83], the data of the financial sector and special treatment (ST) enterprises are often volatile, and estimates based on these data may be unreliable. To prevent the influence of outliers on the results of this paper, the financial industry and ST enterprises are excluded. (2) To eliminate the influence of invalid samples, corporates listed after NEPL are eliminated in DID regression part and all variables are winsorized at the 1% and 99% levels.

Meanwhile, giving that the difference in the ownership backgrounds and industries may bring about disparities in the effects of environmental regulation, we divide ownership backgrounds and industries into different types.

Based on the backgrounds of the major shareholders or actual controllers, our study divides the sample enterprises into two categories, namely, SOEs and non-SOEs [5]. SOEs contain central state-owned enterprises and local state-owned enterprises. The former refers to enterprises whose major shareholders or actual controllers belong to the State-owned Assets Supervision and Administration Commission (SASAC), central state organs or central SOE or institution—the latter refers to enterprises whose major shareholders or actual controllers are from the local SASAC, local governments, departments, or local SOEs or institutions. Non-SOEs include private enterprises, collective enterprises, and public enterprises. Conversely, private enterprises are non-public companies, including individual and private ones, which are characterized by the absence of state-owned capital and non-state holding. Collective enterprises refer to the collective economic system based on workers’ collective ownership of means of production, and whose major way of distribution is in the form of labour (some enterprises implement a distribution that combines labour and capital). Public enterprises are companies without actual controllers.

### 3.2. Variable Definition


(1)Explained variable-corporate financial performance: Our primary measures of a firm’s financial performance are return on assets (*ROA*) and return on equity (*ROE*) based on previous research [83,84,85,86]. Financial performance reflects how well a company generates revenue and manages its assets, liabilities, and the financial interests of its stakeholders. *ROA* is the ratio of the firm’s net income to total assets, which is an indicator of how profitable a company is relative to its total assets. The higher the index is, the better the financial performance. *ROE* is a measure of financial performance calculated by dividing the net income by shareholders’ equity of the firm. It measures the profitability of a business in relation to its equity and can reflect how well a company uses investments to generate earnings growth, known also as net assets or assets minus liabilities. *ROE* is also a positive indicator in measuring a firm’s financial performance.(2)Explanatory variables regarding the implementation of NEPL: According to the paradigm of DID model, we construct two dummy variables, $Treat_{i}$ and $Policy_{t}$, to represent the experimental group variable and time variable, respectively. The HPEs are the experimental group and are denoted as $Treat_{i}=1$, the LPEs are the control group denoted as $Treat_{i}=0$. $Policy_{t}$ is the time dummy variable whose value equals to 1 during and after the implementation of the NEPL, whereas $Policy_{t}=0$ before the implementation of the NEPL. ${\mathrm{Treat}}_{i}\times {\mathrm{Policy}}_{t}$ is the interaction term between the experimental group dummy variable and the policy implementation dummy variable, which reflects the real policy effect.(3)Control variables: To address the issue of missing variables, we select the following controlled variables based on previous research: firm size (Lasset) [83,85], leverage (Alr) [83,84,85], shareholding ratio of the largest shareholder (Fh) [5,53,83], equity balance ratio (Balance) [55], ownership property (State) [5,53], net cash flow per asset (Ttm) [53,54,55], agency cost (Rmana) [53,54,80], growth ability (Inc) [53,55], enterprise age (Age) [53,55]. Table 1 summarises the definitions of the variables.


### 3.3. Econometric Model

DID is a statistical technique widely used in the social sciences. It attempts to mimic an experimental research design using observational data studying the differential effect of a treatment on a treatment group versus a control group in a natural experiment [81]. It calculates the effect of a treatment (i.e., an independent variable) on an outcome (i.e., a dependent variable) by comparing the average change over time in the outcome variable for the treatment group, compared to the average change over time for the control group. The rationale behind using the DID model is that the implementation of the NEPL, on one hand, causes differences between the operating conditions of enterprises before and after, and on the other, causes differences between the performance of HPEs and LPEs. The estimation based on this double difference effectively controls the influence of other synchronous policies, as well as the ex-ante difference between HPEs and LPEs, essentially identifying the causal effect brought by the NEPL.

To study the impact of NEPL on the financial performance of the enterprises, we establish the following models based on [48,49,50,51] to carry out the empirical analysis:(1)$$ROA_{it}=\beta_{0}+\beta_{1}Treat_{i}+\beta_{2}Policy_{t}+\beta_{3}Treat_{i}\times Policy_{t}+\lambda \sum Control_{it}+\alpha_{i}+\gamma_{t}+\varepsilon_{it},$$
(2)$$ROE_{it}=\beta_{0}+\beta_{1}Treat_{i}+\beta_{2}Policy_{t}+\beta_{3}Treat_{i}\times Policy_{t}+\lambda \sum Control_{it}+\alpha_{i}+\gamma_{t}+\varepsilon_{it}.$$

Equations (1) and (2) consider the impacts of the NEPL on *ROA* and *ROE*, respectively. In Equation (1), $ROA_{it}$ is the ratio of the *i*-th firm’s net income to total assets in the period *t*; In Equation (2), $ROE_{it}$ is the ratio of *i*-th firm’s net income to book value of equity in the period *t*. $Treat_{i}$ represents whether the *i*-th corporate is HPEs; $Policy_{t}$ defines whether the NEPL is implemented in period *t*. ${\mathrm{Treat}}_{i}\times {\mathrm{Policy}}_{t}$ is the interaction term between $Treat_{i}$ and $Policy_{t}$. $Control_{it}$ is the control vector. $\alpha_{i}$ and $\gamma_{t}$ are the fixed effects of industry and year, respectively, while $\varepsilon_{it}$ is the random disturbance term.

As for the meaning of the coefficients, In Equations (1) and (2), *β*_1_ and *β*_2_ are the coefficients of the experimental group dummy variables and the time dummy variables, respectively. *β*_1_ characterises the difference between the experimental group and the control group—a difference that exists even without the NEPL. *β*_2_ describes the difference between the periods before and after the implementation of NEPL, that is, even if the NEPL is not performed, the temporal trend exists. *β*_3_ reflects the net effect of policy implementation, which is also the most concerned when we use the DID model. λ is the coefficient matrix of the control variables.

When analyzing the regression results, the point of interest in this paper is the coefficient of *Treat_i_* × *Policy_t_*. If the coefficient of *Treat_i_* × *Policy_t_* is significantly negative, it indicates that the introduction of NEPL has indeed reduced the financial performance of HPEs, and therefore Hypothesis H1 is verified.

## 4. Empirical Analysis, Results and Discussion

### 4.1. Descriptive Statistical Analysis

Table 2 presents summary statistics of all variables. For the whole sample, the mean value of *ROA* is 1.0425, with the minimum and the maximum value of −4.6373 and 7.1506, respectively. The other dependent variable is *ROE*, with a mean value of 1.6615, and the minimum and maximum values of −16.0114 and 13.8810, respectively. The variables *Treat* × *Policy*, *Treat*, *Policy*, and State are binary variables, with values of 0 and 1. The variation range of Alr and Rmana is relatively large, that is, 3.8790 to 93.4627% and 0.7877 to 87.4977% respectively, whereas the variation range of other control variables is relatively small.

### 4.2. Benchmark Regression Analysis

To establish whether the introduction of NEPL has an impact on the financial performance of enterprises, we use the DID model for analysis. Table 3 shows the results of the DID test.

Columns (1) and (2) show the regression results with *ROA* as the explained variable. The former is a simple regression with no control variables, whereas the latter is a dual fixed effect DID model controlling both industry effect and year effect. Similarly, columns (3) and (4) show the regression results with *ROE* as the explained variable. Columns (1) and (3) present the regression results without considering any control variables and only serve as a baseline. Therefore, we mainly focus on the results of columns (2) and (4). 

As can be seen from columns (2) and (4), whether *ROA* or *ROE* are taken as the explained variables, the coefficient of *Treat* × *Policy* is significantly negative (−0.1406 *** and −0.2918 ***, respectively) at the significance level of 1%. More precisely, the coefficient of *Treat* is negative (−0.6204 *** and −1.1663 *), indicating that the average *ROA*/*ROE* of HPEs is lower than that of LPEs. This may be related to the fact that HPEs need to spend larger proportions of their profits on environmental protection compared to LPEs, thus reducing their return rate [54]. The coefficient of *Policy* is negative (−0.1708 *** and −0.3222 ***, respectively), which indicates that, on average, the financial performance of enterprises is decreasing after the implementation of NEPL. Combined with the coefficient of *Treat* × *Policy*, it can be seen that the introduction of NEPL has a significantly negative impact on the financial performance of HPEs. Thus, the empirical results support Hypothesis 1.

In addition, the regression results of the controlled variables are in line with the expectations. Among them, the higher the Lasset, Fh, Balance, Ttm, Inc, and Age, the better the financial performance of the company. This may be because these indicators reflect a company’s profitability to some extent, thus the financial performance can be identified by investors.

### 4.3. The Analysis of Heterogeneity

Table 4 reports the results concerning how NEPL affects the financial performance of SOEs and non-SOEs. To be specific, the coefficient of *Treat* × *Policy* was significantly negative for SOEs (−0.2147 *** and −0.4835 ***). Similarly, for non-SOEs, the coefficient of *Treat* × *Policy* was −0.0914 *** and −0.1573 ***, indicating that the introduction of NEPL significantly reduced the performance of HPEs for both state-owned and non-state-owned.

### 4.4. Analysis of Environmental Supervision Intensity

Given that the implementation of NEPL began in January 2015, and environmental supervision began in 2016, we select data from 2015Q1–2019Q3 to exclude the impact of environmental change on the results. Based on the above analysis, we further evaluate the policy effect of environmental supervision intensity (ESI) to understand whether changes in ESI will have an impact on enterprises’ financial performance. Thus, to examine the inverted U-shape effect of environmental supervision, we use the nonlinear FE model to analyse the relationship. The econometric models are constructed as follows:(3)$$ROA_{it}=\beta_{0}+\beta_{1}Intensity_{t}+\beta_{2}Intensity_{t}^{2}+\lambda \sum Control_{it}+\alpha_{i}+\gamma_{t}+\varepsilon_{it},$$
(4)$$ROE_{it}=\beta_{0}+\beta_{1}Intensity_{t}+\beta_{2}Intensity_{t}^{2}+\lambda \sum Control_{it}+\alpha_{i}+\gamma_{t}+\varepsilon_{it}.$$

The core independent variables $Intensity_{t}$ and $Intensity_{t}^{2}$ represent the intensity of environmental supervision in the *t*-th year and its quadratic term, respectively. Intensity=0 means the province has not been supervised by central government, and Intensity = 1 means that the province has been supervised once, whereas Intensity=2 means that the province has been supervised twice. It is important to clarify that *Intensity* is assumed to be a continuous variable in this paper. Since 2016, in addition to the central environmental supervision, local governments at all levels have also implemented various local environmental supervision. That is to say, environmental supervision is a continuous behaviour in the time dimension, so *Intensity* is a continuous variable. After the third quarter of 2019, there may be more central or local inspections in the future, but we can only observe 0, 1 or 2 at the moment. Meanwhile, since the inspection intensity at the local level is usually weaker than that at the central level, in order to simplify the analysis, the paper still uses 0, 1, 2 to represent the change of ESI from 2016 to 2019. The other variables have the same meaning as above.

In terms of model selection, for the panel data of listed companies, the Hausman test is performed on all benchmark regressions, and the test results reject the null hypothesis of the random-effects model. Thus, we use the FE model to evaluate the policy effect of ESI, and Table 5 reports the impact of ESI on enterprises’ financial performance.

Column (1) and column (4) of Table 5 show the regression results for the whole sample. Specifically, the coefficient of *Intensity*^2^ and *intensity* is −0.0935 *** and 0.1222 *** respectively with *ROA* as the explained variable, and both passed 1% significance test. Similarly, the coefficient of *Intensity*^2^ is −0.1991 *** and 0.1947 *** for *intensity* with *ROE* as the explained variable, and both passed 1% significance test as well. Thus, it can be deduced that there is a significant inverted U-shape relationship between ESI and the financial performance of enterprises. Columns (2) and (5) illustrate the regression results for non-SOEs, whereas columns (3) and (6) refer to SOEs, both of which reflect an inverted U-shape relationship, that is, with the intensification of environmental supervision, the performance of non-SOEs and that of SOEs both increase at first, and then begin to decline.

More specifically, by comparing columns (2) and (3), it can be found that the absolute value of quadratic term coefficient in non-SOEs (0.0800) is less than that of SOEs (0.1054), indicating that the financial performance of SOEs is more sensitive to environmental regulation, and the inverted “U-shape” curve is steeper. In addition, through a simple calculation, we can know that the inflection point value of non-SOEs is 0.399, which is less than 0.958 of SOEs, indicating that the financial performance of non-SOEs has a weaker tolerance to environmental regulation, that is, an earlier inflection point in the inverted “U-shape” curve. Columns (4) and (5) exhibit the same characteristics.

Furthermore, we examine the influence of ESI on HPEs and LPEs of SOEs, the empirical results are presented in Table 6.

Columns (1) and (2) show the effect of ESI on corporate financial performance of state-owned LPEs and HPEs with *ROA* as the explained variable. In column (1), the coefficient of *Intensity*^2^ is −0.0872 ***, which means that there is an inverted U-shape relationship between ESI and corporate financial performance of LPEs. And in column (2), since the coefficient of *Intensity*^2^ is −0.1569 ***, similar relationship can be identified for HPEs. Columns (4) and (5) exhibit the same feature, with *ROE* as the explained variable.

We also examine the influence of ESI on HPEs and LPEs of non-SOEs. The empirical results are presented in Table 7.

As illustrated in Table 7, strengthening environmental supervision, the performance of non-state-owned HPEs and LPEs shows similar characteristics to that of SOEs. That is, the increase of ESI enhances the performance of non-state-owned HPEs and LPEs at first and then reduces it. Considering this alongside Table 6 and Table 7, it can be found that there is a significant inverted U-shaped relationship between ESI and HPEs as well as with LPEs, both for state-owned enterprises and non-state-owned enterprises. Therefore, our Hypothesis 2 is verified.

In addition, by comparing Table 6 and Table 7, it can be inferred that, for LPEs, the absolute value of the quadratic term coefficient in non-SOEs (0.0670/0.1216) is less than that of SOEs (0.0872/0.2189). Moreover, the inflection point value of non-SOEs is 0.474 (−0.046), which is also lower than that of SOEs (0.924/0.797), indicating that the financial performance of state-owned LPEs is more sensitive and tolerant to environmental regulation than that of non-state-owned LPEs. Similarly, for HPEs, the absolute value of the coefficient of *Intensity*^2^ in non-SOEs (0.1060/0.2486) is less than that of SOEs (0.1569/0.3366). Moreover, the inflection point value of non-SOEs is 0.133 (0.219), which is also lower than that of SOEs (0.869/0.816), demonstrating that the financial performance of state-owned HPEs is more sensitive and tolerant to environmental regulation than that of non-state-owned HPEs. Therefore, Hypothesis 3 is also verified.

Based on the results reported in Table 5, Table 6 and Table 7, it can be concluded that there is a remarkable inverted “U-shape” relationship between ESI and the financial performance of HPEs and LPEs, regardless of SOEs or non-SOEs. In addition, no matter HPEs or LPEs, enterprises with state-owned background show higher sensitivity and tolerance when subjected to environmental regulation.

## 5. Conclusions

Taking Chinese A-share listed companies as samples, this paper explores the relationship between environmental regulation, namely NEPL and ESI in this paper, and corporate financial performance under the background of industry heterogeneity and ownership heterogeneity, using DID model and nonlinear FE model respectively. We found that the implementation of NEPL significantly reduced the financial performance of both state-owned and non-state-owned HPEs. Further analysis of ESI showed that there is a significant inverted U-shape relationship between ESI and the financial performance of HPEs and LPEs, regardless of SOEs or non-SOEs. Moreover, the financial performance of SOEs is more sensitive and tolerant to environmental regulation than that of non-SOEs, reflected in a steeper inverted U-shaped curve and a slower inflection point.

Environmental protection and a sustainable governance system have increasingly become key factors that must be considered in business operations. To better implement a sustainable development policy and further promote the construction of China’s ecological civilization, three policy recommendations, based on the above research, are put forward.

First, environmental problems need to attract extensive attention from countries all over the world, including China. Despite various policies and initiatives by governments and NGOs for the protection of the environment to achieve sustainable development of human society, it is still common for many countries and enterprises to ignore policies and decrees based on their own interests. The practice of trading high pollution for high returns remains widespread. Environmental degradation, characterized by air pollution, can cause significant increases in mortality and morbidity rates due to respiratory disorders, particularly among children [87,88]. Also, it has led to a series of problems such as global warming, frequent occurrence of extreme weather, and severe destruction of biodiversity, which may be irreversible. Therefore, protecting the environment should not only be a slogan; the seriousness of environmental problems and the urgency of solving them should be promptly acknowledged. The international community should strengthen environmental protection and encourage more countries to join the global environmental agreement to jointly promote the establishment of a global environmental governance system. Similarly, at a national level, scientific and reasonable environmental protection policies should be formulated to increase the illegal costs of environmental pollution to internalize the externalities of environmental pollution.

Secondly, firm size and political connection should be considered when formulating environmental policies. Uniform aggregate environment standards can have a devastating impact on the operations of small businesses. Formulating differentiated environmental constraints, and setting environmental emission standards in proportion rather than in total, so as to protect SMEs from excessive pressure of environmental regulation. Also, in the implementation of environmental policies, the negative effects of political connections should be fully considered. Establishing a more transparent disclosure system of environmental regulation and taking the third-party intermediary as the executor of policies may alleviate the inequality of policy implementation to some extent.

Finally, sustainable development, as a treaty to be observed by all countries in the world, should be earnestly implemented. Sustainable development means meeting the needs of the present without compromising the ability of future generations to meet their own needs [89]. Although there is now a general consensus that environmental protection must be strengthened to achieve sustainable economic and social development, the implementation of this idea is lagging. For example, despite the adoption of 17 Sustainable Development Goals by all UN member states in 2015, the failure of the Copenhagen climate summit to produce a legally binding text and the withdrawal of the U.S. from the Paris Agreement all indicate that global enforcement of such protection is yet to be accomplished. To cope with the increasingly pressing environmental, political, and economic challenges facing the world, countries should abandon the concept of maximizing their own interests, take the initiative to assume their respective responsibilities in environmental governance, and earnestly implement the concept of sustainable development. This is the only way to guarantee long-term development.

The limitation of this paper is that the study on the relationship between ESI and corporate financial performance is based on the setting of parameter model rather than from a non-parametric perspective. Under the setting of parameter model, the correct relationship identified between the two is an inverted U-shape, that is, our current research conclusion. However, if analyzed from a non-parametric perspective, more complex relationships may be identified. Therefore, in the future, we hope to conduct a more in-depth study of this problem from a non-parametric perspective, so as to find some new relations between the two.

## Figures and Tables

**Table 1 ijerph-17-03828-t001:** Summary of the definitions of variables.

Variables	Definition
*ROA*	Ratio of the firm’s net income to total assets
*ROE*	Ratio of net income to book value of equity
*Treat* × *Policy*	The interaction item of the group dummy variable and the time dummy variable
*Treat*	If the company is high pollution, *Treat* = 1, otherwise, *Treat* = 0.
*Policy*	After the implementation of NEPL, *Policy* = 1, Before the implementation of NEPL, *Policy* = 0.
State	If the company is state-owned, State = 1, otherwise, State = 0.
Lasset	Natural log of the firm’s total asset
Alr	Ratio of the firm’s long-term debt to total assets
Fh	Shareholding ratio of the largest shareholder
Balance	Shareholding ratio of the second to tenth shareholders
Ttm	The ratio of cash inflows minus outflows to total assets
Rmana	Ratio of administrative expenses to operating revenue
Inc	Revenue growth rate
Age	Enterprise age

**Table 2 ijerph-17-03828-t002:** Summary statistics of the variables.

Variables	Observations	Mean	Std Dev	Min	Max
*ROA*	41,700	1.0425	1.6996	−4.6373	7.1506
*ROE*	41,457	1.6615	3.6379	−16.0114	13.8810
*Treat* × *Policy*	42,498	0.1465	0.3537	0	1
*Treat*	42,498	0.2931	0.4552	0	1
*Policy*	42,498	0.5000	0.5000	0	1
State	42,498	0.3892	0.4876	0	1
Lasset	41,793	8.1516	1.2858	5.6136	12.0641
Alr	41,774	42.7360	22.2072	3.8790	93.4627
Fh	41,892	36.0821	15.3286	8.9500	76.9500
Balance	37,715	21.9215	13.1572	1.9400	55.0200
Ttm	41,791	0.2455	9.9732	−27.3535	41.4743
Rmana	41,675	11.7522	12.4602	0.7877	87.4977
Inc	39,318	0.0332	0.4726	−1.5833	1.7331
Age	42,498	9.5270	6.8259	−2.0000	26.0000

**Table 3 ijerph-17-03828-t003:** The impact of the NEPL on enterprises’ financial performance.

Variables	*ROA*	*ROA*	*ROE*	*ROE*
	(1)	(2)	(3)	(4)
*Treat* × *Policy*	−0.1157 ***	−0.1406 ***	−0.2793 ***	−0.2918 ***
	(0.0378)	(0.0324)	(0.0822)	(0.0773)
*Treat*	0.0486 *	−0.6204 ***	−0.1974 ***	−1.1663 ***
	(0.0269)	(0.1789)	(0.0576)	(0.2580)
*Policy*	−0.0751 ***	−0.1708 ***	−0.1617 ***	−0.3222 ***
	(0.0193)	(0.0406)	(0.0409)	(0.0907)
Lasset		0.1779 ***		0.5216 ***
		(0.0098)		(0.0260)
Alr		−0.0281 ***		−0.0411 ***
		(0.0006)		(0.0017)
Fh		0.0099 ***		0.0178 ***
		(0.0006)		(0.0015)
Balance		0.0129 ***		0.0227 ***
		(0.0008)		(0.0017)
Ttm		0.0182 ***		0.0313 ***
		(0.0009)		(0.0018)
Rmana		−0.0288 ***		−0.0623 ***
		(0.0012)		(0.0027)
Inc		0.7326 ***		1.4699 ***
		(0.0231)		(0.0537)
Age		0.0033 **		0.0075 **
		(0.0016)		(0.0036)
Constant	1.0837 ***	0.7355 ***	1.8439 ***	−0.7991 ***
	(0.0138)	(0.1068)	(0.0291)	(0.2436)
Industry	No	Yes	No	Yes
Year	No	Yes	No	Yes
Observations	41,700	36,921	41,457	36,743
R-squared	0.0013	0.3166	0.0032	0.2304

Note: The symbols ***, ** and * denote significance at the 1%, 5%, and 10% levels, respectively. The values in parentheses are robust standard errors.

**Table 4 ijerph-17-03828-t004:** Impacts of NEPL on the performance of SOEs and Non-SOEs.

Variables	SOEs	Non-SOEs	SOE	Non-SOEs
	*ROA*	*ROA*	*ROE*	*ROE*
	(1)	(2)	(3)	(4)
*Treat* × *Policy*	−0.2147 ***	−0.0914 **	−0.4835 ***	−0.1573 *
	(0.0479)	(0.0424)	(0.1295)	(0.0918)
*Treat*	−0.4754 **	−0.7491 **	−1.0658 ***	−1.2295 **
	(0.1975)	(0.3111)	(0.2767)	(0.5342)
*Policy*	−0.3155 ***	−0.1216 **	−0.6786 ***	−0.2171 **
	(0.0604)	(0.0538)	(0.1533)	(0.1091)
Constant	0.9603 ***	0.4702 **	0.3347	−2.0654 ***
	(0.1265)	(0.2271)	(0.3306)	(0.4759)
Controls	Yes	Yes	Yes	Yes
Industry	Yes	Yes	Yes	Yes
Year	Yes	Yes	Yes	Yes
Observations	14,720	22,201	14,655	22,088
R-squared	0.3534	0.3189	0.2671	0.2432

Note: The symbols ***, ** and * denote significance at the 1%, 5%, and 10% levels, respectively. The values in parentheses are robust standard errors.

**Table 5 ijerph-17-03828-t005:** The Influence of ESI on Enterprise Performance.

Variables	Overall	Non-SOEs	SOEs	Overall	Non-SOEs	SOEs
	*ROA*	*ROA*	*ROA*	*ROE*	*ROE*	*ROE*
	(1)	(2)	(3)	(4)	(5)	(6)
*Intensity*	0.1222 ***	0.0639	0.2019 ***	0.1947 ***	0.0406	0.4320 ***
	(0.0323)	(0.0396)	(0.0551)	(0.0716)	(0.0809)	(0.1389)
${\mathrm{Intensity}}^{2}$	−0.0935 ***	−0.0800 ***	−0.1054 ***	−0.1991 ***	−0.1605 ***	−0.2500 ***
	(0.0142)	(0.0176)	(0.0236)	(0.0311)	(0.0361)	(0.0584)
Constant	0.9964 ***	0.6680	0.8248	−1.1436	−1.8075 *	−0.8036
	(0.3549)	(0.4354)	(0.6864)	(0.8748)	(0.9846)	(1.9523)
Controls	Yes	Yes	Yes	Yes	Yes	Yes
Observations	56,744	38,332	18,412	56,636	38,283	18,353
R-squared	0.1670	0.1753	0.1626	0.1269	0.1333	0.1260

Note: The symbols *** and * denote significance at the 1% and 10% levels, respectively. The values in parentheses are robust standard errors.

**Table 6 ijerph-17-03828-t006:** The impact of ESI on the performance of HPEs and LPEs of SOEs.

Variables	LPEs	HPEs	LPEs	HPEs
	*ROA*	*ROA*	*ROE*	*ROE*
	(1)	(2)	(3)	(4)
*Intensity*	0.1611 ***	0.2727 **	0.3489 **	0.5490 *
	(0.0598)	(0.1195)	(0.1532)	(0.2971)
${\mathrm{Intensity}}^{2}$	−0.0872 ***	−0.1569 ***	−0.2189 ***	−0.3366 ***
	(0.0248)	(0.0542)	(0.0638)	(0.1281)
Constant	0.3895	1.6732	−2.5402	3.4976
	(0.7487)	(1.2761)	(2.1949)	(3.6169)
Controls	Yes	Yes	Yes	Yes
Observations	12,980	5,432	12,940	5,413
R-squared	0.1534	0.2188	0.1252	0.1649

Note: The symbols ***, ** and * denote significance at the 1%, 5%, and 10% levels, respectively. The values in parentheses are robust standard errors.

**Table 7 ijerph-17-03828-t007:** The impact of ESI on the performance of HPEs and LPEs of non-SOEs.

Variables	LPEs	HPEs	LPEs	HPEs
	*ROA*	*ROA*	*ROE*	*ROE*
	(1)	(2)	(3)	(4)
*Intensity*	0.0635	0.0283	−0.0111	0.1090
	(0.0464)	(0.0737)	(0.0957)	(0.1452)
${\mathrm{Intensity}}^{2}$	−0.0670 ***	−0.1060 ***	−0.1216 ***	−0.2486 ***
	(0.0202)	(0.0343)	(0.0427)	(0.0648)
Constant	1.0262 **	−0.8863	−1.3621	−4.2348
	(0.4720)	(0.9208)	(0.9913)	(2.5858)
Controls	Yes	Yes	Yes	Yes
Observations	27,803	10,529	27,767	10,516
R-squared	0.1883	0.1573	0.1419	0.1248

Note: The symbols *** and ** denote significance at the 1% and 5% levels, respectively. The values in parentheses are robust standard errors.

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
