# Peer review of "Promoting or Inhibiting? The Impact of Environmental Regulation on Corporate Financial Performance—An Empirical Analysis Based on China"

_ijerph, 2020, doi:10.3390/ijerph17113828_

Round 1

Reviewer 1 Report

Promoting or Inhibiting? The Impact of Environmental Regulation on Corporate Financial Performance—An Empirical Analysis based on China

Thank you for the opportunity to read and review this paper. The topic of the paper is relevant, but overall, I feel that the paper as currently written lacks sufficient theoretical focus and clarity, or relatedly a strong theoretical contribution. My following comments are intended to be constructive and hope they are helpful to the authors as they move forward with this project.

In the introduction, it is noticeable the lack of motivation that the article has. For this reason, my suggestion to the author is to try to include the research question which help to explain and clarify the lack of previous research that addresses this topic. Authors should clearly demonstrate the contribution of this paper and in what manner it is different from previous papers. The authors underestimate their work by not stating their contribution properly. What exactly is new about your research? The contribution to the literature should be improved.

Where is your theoretical section and hypothesis development.? I was expecting a theoretical background supporting your main hypotheses. A theoretical foundation must be developed, focusing specifically on the core of your topic. The authors allude to previous empirical studies, but don't include hypotheses with a real theoretical support with a solid perspective or approach.

Benchmark Model and Data Resource is clear and correct. However, the discussions of the results should be enriched. The main results obtained in the Table 3-7, should be supported by the theories and the hypotheses proposed.

Conclusions: This section should emphasis the objective and main results presented in this study. Moreover, this section should include limitations and future lines of research.

Good Luck!!!!

Reviewer 2 Report

Manuscript ID: ijerph-809055

Title: Promoting or Inhibiting? The Impact of Environmental Regulation on Corporate Financial Performance—An Empirical Analysis based on China

Comments:

The paper technology presents the relationship between the New Environmental Protection Law (NEPL) launched in China and corporate financial performance. In addition, authors try to find the impact of environmental supervision intensity (ESI) on various enterprises. After a further round of an in-depth review by the authors, I would like to comment on the following academic weaknesses that should be addressed before any consideration for publication. Before any future submission for publication, however, it would be good for the authors to consider the following comments.

Recommendation: Major revision

More specifically,

A major revision is requested before any attempt for publication will most likely require to take into account the following comments.

  • The title is too large even it does reflect the content of the paper.
  • Abstract is well written. However, authors should shorten it giving more concrete and important information to the reader. The aim and the innovation of the paper is totally missing.
  • The part of introduction does not illustrate clearly the initial innovation of the specific study.
  • “Benchmark Model and Data Resource” part has several issues. The most important issue is the order of the presented parts. For instance, authors start presenting the econometric model and thereafter the data and the variables used in their analysis. Authors should present the data that will be used in the empirical analysis, its summary statistics and then to illustrate and define the empirical methodology that they will follow. It is not appealing for the reader to read about the ROA variable for instance, and the definition to exist some pages after this point. Similarly, one can read about the sample division into HPEs and LPEs companies in line 154 and HPEs’ definition exists in line 200.    
  • Regarding the empirical results, authors should clarify some points that are unclear in the analysis. For instance, why models 1 and 3 (table 3) do not consider fixed effects estimators? Authors should estimate Hausman test and decide which model fits their model. In equations 3 and 4, intensity is not a quantitative variable as it takes only the values of 0, 1 and 2; thus, please explain what exactly is the square term of intensity. Is it possible to find a threshold or it shows only the shape that authors try to find? Also, what is its interpretation for SOEs with intensity equals to 0 or 1?
  • Conclusion part is very large. Authors should shorten it making policy implications more compact and informative. Limitations and clear further research are missing.

Reviewer 3 Report

Very interesting article. The right methodology of research. The research question is clearly stated. The theoretical framework are creative. The research question is explored in a way that is creative and important to the discipline. The methodology is clearly explained. The empirical data, quantitative are analyzed in appropriate ways, and written up in ways that are easy to understand. The study conclusions supported are by the analysis. The analysis of adequately address the issues raised by the framework.

Round 2

Reviewer 1 Report

I appreciate the opportunity to read the paper entitled “Promoting or Inhibiting? The Impact of Environmental Regulation on Corporate Financial Performance—An Empirical Analysis based on China”. I read the manuscript with great attention. I have observed that this manuscrit has improved after the suggestions. So, I recommend the publication in the International Journal of Environmental Research and Public Health

Reviewer 2 Report

After a second review of the updated manuscript, I could say that authors made all required clarifications and the quality of the paper has been highly increased.

This manuscript is a resubmission of an earlier submission. The following is a list of the peer review reports and author responses from that submission.

Round 1

Reviewer 1 Report

This paper joins a series of research work aimed at exploring the link between environmental regulation and corporate financial performance. The contribution claimed by the authors include the impact of environmental supervision intensity and heterogeneity in terms of ownership backgrounds.

The paper is generally well written. However, it suffers from critical literature review, structuring and in terms of good contribution as highlighted in the following points.

The most important limitation is the absence of critical literature review. There is no literature review section to bring out the contributions strongly. Since this is a well-researched field, it is absolutely vital that a research gap is identified via a critical literature review. For example, relevant papers that looked at the impact of industrial heterogeneity in this line of research are not included in the paper (e.g., Ruiqian Li and Ramakrishnan Ramanathan (2018), “Impacts of Industrial Heterogeneity and Technical Innovation on the Relationship between Environmental Performance and Financial Performance”, Sustainability, 10(5), 1653.) The literature in the area of environmental supervision intensity should also be critically reviewed to support the analysis and results. There is complete absence of use of theory to link to the ideas presented in the article. It is important to argue why and how the results are valid by linking to some theoretical framework. The paper could be better structured. I am not sure why Section 3.3 is titled Further Analysis while it only focuses on ESI.

The paper requires a major review before it can be accepted.

Reviewer 2 Report

Page 1 line 20…with the exception of non-SOE low-pollution…

What is non-SOE?

Page 1 line 27 to page 3 line 103 must minor revision, such as page 1 line 35 European Union must be followed by (UK), … page 2 line 52-57: …According to the study of [14–22],…However, [23] believed…literatures presentation content and method must be revised.

(There are many similar situations in line 52-103.)

Although it has been shown that this article has three major contributions, can you explain further why this method is needed. line 130 ROA and ROE must be revised to return on assets (ROA) and return on equity (ROE), and then line 131-132 delete return on assets and return on equity only use ROA and ROE. Line 173 what is ST? Table 3 is not shown in the text (line 190-197). Table 3 column (1) and (2), (3) and (4) the difference between should be explained. Why are there such obvious differences in 0.0277 and -0.119*** of Policy, there should be some management implications here. The difference results of Table 3 should be more specifically explained in text.

Reviewer 3 Report

The paper presents a topical issue, found on the public agenda of most states, but also different international bodies. At the same time, this topic has also attracted researchers’ attention, who have addressed the issue of the impact of environmental regulations in numerous scientific papers.

The sample of companies under study is relevant to the proposed objectives, as well as the period for which the data were collected and processed. The data processing methodology is appropriate to the proposed objectives.

I consider that the paper can be improved in the “Conclusions” section by presenting the economic significance of the causal links identified between environmental regulation and the financial performance of the companies, as well as the causes that have led to a negative relationship between them.
